# Asymmetric Carrier Divider with an Irregular RF Phase on DD-MZ Modulator for Eliminating Dispersion Power Fading in RoF Communication

**Gunawan Wibisono [1],\*, Febrizal Ujang [1], Teguh Firmansyah [2] and Purnomo S. Priambodo [1]**

1   Department of Electrical Engineering, Universitas Indonesia, Jawa Barat 16424, Indonesia;
    febrizal@ui.ac.id (F.U.); p.s.priambodo@ieee.org (P.S.P.)
2   Department of Electrical Engineering, Universitas Sultan Ageng Tirtayasa, Cilegon 42435, Indonesia;
    teguhfirmansyah@untirta.ac.id
\*   Correspondence: gunawan@eng.ui.ac.id

**Abstract:** The main problem of intensity modulation (IM) in radio-over-fiber (RoF) communication is dispersion power fading (DPF), which occurs when the signal is transmitted through a dispersive link that causes a sideband cancelation effect. The DPF level of the RoF link is determined by the deviation factor (DF). The optical single-sideband (OSSB) modulation scheme, which is generated by driving one of the dual-drive Mach–Zehnder modulators (DD-MZMs), is usually used to overcome DPF. The DF value of OSSB modulation at modulation index $m = 0.1$ increases from 0.008 to 0.930 at $m = 1$. It can be said that this method is only effective at reducing DF at low $m$. However, as well-known information of the DD-MZM system, high-efficiency optic–electric conversions can be obtained at high $m$ values, but DF will increase. Therefore, reducing the DPF value for high $m \geq 0.1$ is interesting. It is known that in wireless communication, to reduce the impact of fading, direct signals are amplified and signals with irregular phases are used. Moreover, this paper proposes the DD-MZM with an asymmetric carrier divider as a direct signal and combines it with an irregular radio frequency (RF) phase to reduce the DPF at high $m$. The carrier that is generated by laser diode (LD) power ($P_{IN}$) is divided asymmetrically as power modulation ($P_{DD-MZM}$) and carrier arm (CA) power ($P_{CA}$). Furthermore, the minimum DF is obtained when the $P_{IN}$ is separated as 75% for $P_{CA}$ and 25% for $P_{DD-MZM}$ with an irregular RF signal of $\theta = 48°$ and a bias point value of $\gamma = 3/4$. As a result, with the same power as OSSB, this proposed structure produces DF at $m = 0.1$ and $m = 1$ with values of 0.008 and 0.03, or it can reduce DF of 96.7% at $m = 1$. The mathematical model and simulation model have very good agreement, which validates the proposed method.

**Keywords:** chromatic dispersion; dispersion power fading; radio-over-fiber; DD-MZM; DD-MZM with CA; deviation factor

## 1. Introduction

A system that can transmit radio frequency (RF) signals ($X_{TX}(t)$) through optical fiber used to support wireless communication services, known as radio-over-fiber (RoF), has been developed recently. The transmission is completed by modulating the optical source ($E_{in}(t)$) using $X_{TX}(t)$, which is transmitted using the electro-optic (E/O) converter located in the central office (CO). At the receiver, the RF signal is recovered using an optoelectric (O/E) converter located in the radio access point (RAP). Recovered RF signals ($X_{rec}(t)$) are then transmitted wirelessly from RAP to the mobile station (MS). The conversion of RF signals to optics can be performed by modulating the optical source directly or modulating the optical carrier externally. Either direct or external modulation can be used to modulate the intensity (amplitude) or phase of the optical carrier. The recovery of the RF signal at the receiver on

intensity modulation (IM) is recognized as direct detection (DD), whereas in phase modulation, it is known as coherent detection [1].

The IM on the RoF link produces an optical signal with a double sideband spectrum. The optical double sideband (ODSB) is an optical signal with a spectrum consisting of an optical carrier and upper and lower sidebands located around the optical carrier. When the ODSB signal is transmitted through a fiber link, the chromatic dispersion of the fiber causes the sideband and optical carrier to propagate at different speeds. This leads to the modulated signal at the receiver ($E_{RX}(t)$) experiencing a different phase between the sideband component and optical carrier by $\phi$. The proportion of $\phi$ follows the length of the fiber ($L$), the frequency of the RF signal ($f_m$) and the wavelength ($\lambda_c$) used. The phase difference between the sideband and optical carrier causes the O/E process to generate two identical RF signals but with a different phase of $2\phi$, resulting in constructive and deconstructive interference on the recovered RF signal. Destructive interference causes a reduction in recovered RF signal power, which is known as dispersion power fading (DPF). If $\phi = \pi/2$, the sideband cancelation effect occurs, which causes a massive loss of power or deep fade [2].

There are several methods for overcoming the DPF, such as carrier phase shift (CPS) [3–8]. The CPS method uses a carrier phase shift of the $E_{TX}(t)$ signal at the transmitter. The phase of the carrier ($\phi$) is adjusted such that $\phi$ at the receiver is zero. However, this method has disadvantages because $\phi$ is different due to the change in $L$, $f_m$, and $\lambda_c$. Thus, it is always necessary to reset $\phi$ every system change, and it is not a convenient or robust solution. Furthermore, optical carrier suppression (OCS) was proposed by [9–16]. The OCS modulation system is modulated with upper and lower sidebands but without carrier components. At the OCS, $X_{rec}(t)$ is generated from the beating upper sideband and lower sideband with a distance of $2f_m$, so the frequency of $X_{rec}(t)$ is $2f_m$. However, this method has drawbacks, such as requiring additional devices, such as downconverters, to convert $X_{rec}(t)$ frequency to its initial value. Machine learning can also be used to overcome DPF. This method has been successfully implemented for short-range transmission [17,18]. However, it also has some disadvantages, one of which is a complicated configuration on the receiver side.

The reduction in DPF can be overcome using optical single-sideband (OSSB) modulation. OSSB modulation has advantages such as not requiring frequency translation [19] or inconvenient phase receiver adjustment. OSSB modulation scheme can be generated by filtering one of the sideband modulated optical signals [20]. This method, however, highly dependents on the optical wavelength. The OSSB modulation scheme can be produced with DD-MZM by driving one of its arms using the RF signal that is given a bias voltage 1/2 of the switching voltage and driving another arm using a signal with a regular phase ($\theta$) of 90° [15,21–24]. DD-MZM is an E/O converter frequently used in RoF links due to its ability to generate various optical schemes with just a simple configuration.

The mathematical model of OSSB modulation is usually developed based on the Bessel function [15]. Based on this function, the low DPF at OSSB can be obtained if the system generates the carrier signal and +1st order sideband only. To produce this condition, the OSSB modulation must be operated at low $m \leq 0.1$ because higher $m$ will generate more sideband orders. A higher number of sideband orders will automatically increase the DPF. However, the OSSB is only effective at low $m \leq 0.1$. However, to increase the efficiency of optic–electric conversions at the OSSB, a high $m$ value is needed [22]. Therefore, it will be interesting to reduce the DPF value at high $m = 1$.

In wireless communication, increasing the direct signal power [25] and using irregular phases can reduce the effect of fading. Then, the concept of suppressing fading in wireless communication can be applied to RoF communication to suppress the effect of DPF. As a novelty, this paper proposes an asymmetric carrier divider as a direct signal with an irregular RF phase on a DD-MZ modulator for eliminating DPF, as shown in Figure 1. In detail, the optical source generated using a laser diode (LD), $E_{in}(t)$ is divided using an optical splitter (OS) with asymmetrical values such as power modulation ($P_{DD-MZM}$) and carrier arm (CA) power ($P_{CA}$). The signal at $P_{DD-MZM}$ is modulated using DD-MZM, and the modulated signal ($E_{DD-MZM}(t)$) is generated. The signal at the CA is added to $E_{DD-MZM}(t)$ using an optical combiner (OC), which produces the transmitted signal ($E_{TX}(t)$). The $E_{TX}(t)$ is transmitted

using fiber link $H(f)$ and produces received signal ($E_{RX}(t)$). The $E_{RX}(t)$ is detected and recovered using a photodetector (PD) and produces $X_{rec}(t)$. This proposed method is called an asymmetric carrier divider system.

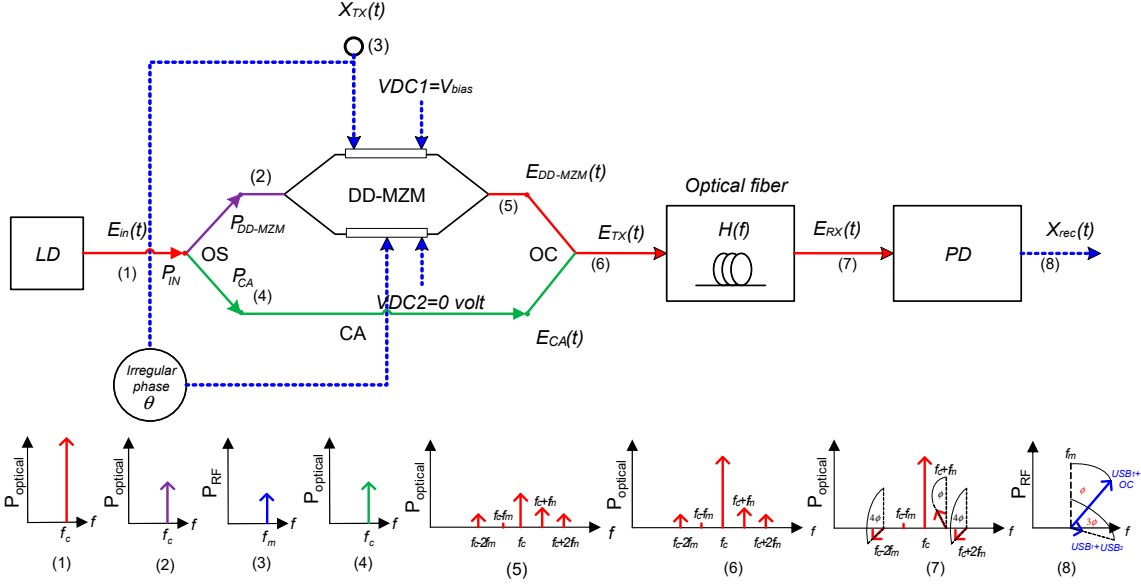

**Figure 1.** The proposed asymmetric carrier divider with irregular RF phase on DD-MZ modulator.

Furthermore, to further reduce the DPF value that is caused by multi-sideband order, this paper also proposes the irregular RF phase. Therefore, the optimum $E_{DD-MZM}(t)$ spectrum can be generated by adjusting the DD-MZM arm to produce a low DPF. The DPF level is determined using the deviation factor (DF). DF is the quantity of recovered RF signal power variation at the receiver that is measured over a certain range of fiber lengths. The smaller the DF is, the lower the DPF level or the smaller the effect of chromatic dispersion towards the RoF link.

Therefore, we combine the asymmetric carrier divider system with the irregular RF phase on the DD-MZ modulator. Therefore, the combination of the optimum value of the power ratio and the optimum value of the irregular phase will reduce DF at RoF communication significantly.

Furthermore, the significant contributions of this paper are as follows:

1. The laser diode (LD) power ($P_{IN}$) carrier is divided asymmetrically as the power modulation ($P_{DD-MZM}$) and carrier arm power ($P_{CA}$). The power $P_{CA}$ is used to compensate for the power of the carrier of the optical field of the DD-MZM output, which is reduced due to increasing $m$.
2. The RF signal with an irregular phase ($\theta$) was applied to the DD-MZ modulator. Therefore, the optimum $E_{DD-MZM}(t)$ spectrum can be generated by adjusting the DD-MZM arm.
3. The minimum DF is obtained when the $P_{IN}$ is separated as 75% for $P_{CA}$ and 25% for $P_{DD-MZM}$ with an irregular RF signal of $\theta = 48°$ and bias point value of $\gamma = 3/4$. This proposed structure produces DF at $m = 0.1$ and $m = 1$ value are 0.008 and 0.03, or it reduced DF of 96.7% compared to OSSB.
4. The proposed system was successfully applied without additional power or filter, and the additional power or filter managed to increase the cost and complexity of RAP. Moreover, $\theta$ is independent of fiber length. Our proposed model is applied for single fiber, not for routing scenarios with many nodes.
5. The proposed mathematical model of the system is developed based on the Bessel function. This model can be used to evaluate all the parameters. Furthermore, the model is validated using simulation, and it has very good agreement, which validates the proposed method.

6. Nevertheless, the current paper has some limitations: (1) We used the comparison between numerical and simulation models without experiment due to the unavailability of devices, but this model was successfully verified and had a good agreement; (2) Our proposed method was focused on optical signals modulated by RF sine waves [22,26–29]; (3) To eliminate the dispersion, we focused on the received RF power; (4) We assumed that the optic arm length is less than 1 m; therefore, the phase did not change significantly; (5) In this study, we investigated up to 16 scenarios, and the DD-MZM modulator was used with the best performance in the C4 scenario; and (6) As a consideration, the paper's proposed method focused not only on asymmetric power divider but also on the optimal phase. Therefore, the minimum DF was obtained by combining the irregular phase and the power divider to reduce DF by 96.7%. We think that the global optimum DF results will also be similar to the proposed case in this scenario. Moreover, we also must consider the availability of the optical divider in the market.

## 2. Optical Field of DD-MZM Output

### 2.1. DD-MZM with Basic Configuration

Mach–Zehnder modulator (MZM) is an external IM generally used on fiber optic links. MZM can be driven on one or both arms. MZM that is driven on both arms is known as dual-drive MZM (DD-MZM). With the configuration illustrated in Figure 2, DD-MZM can produce several modulation schemes shapes. $E_{in}(t)$ generated by the laser diode (LD) is equally divided into both arms (upper and lower arms). The MZM upper arm is driven by the RF signal ($X_{TX}(t)$) and is given an additional DC voltage of $V_{bias}$. The lower arm of the MZM is driven by the same $X_{TX}(t)$ signal, but its phase is shifted using an electrical phase shifter (EPS) of $\theta$ without an additional DC voltage.

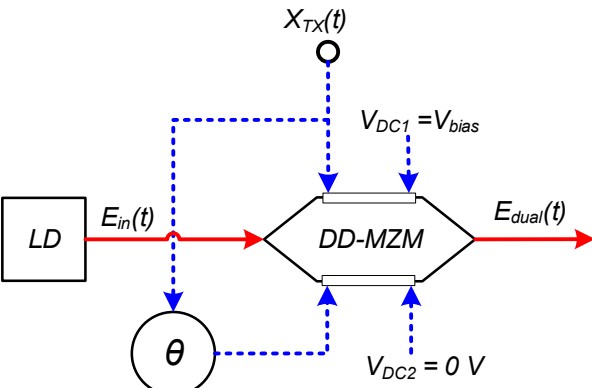

**Figure 2.** Basic configuration of DD-MZM to generate various modulation schemes.

Assuming that the extinction ratio (ER) of DD-MZM is extremely high, and the optical signal has the same polarization at every point, the optical field of DD-MZM output $E_{dual}(t)$ can be approximated by [30]

$$E_{dual}(t) \approx \frac{1}{2}\sqrt{2P_{in}}e^{j2\pi f_c t}\left\{e^{\left(j\pi\frac{V_{up}(t)}{V_\pi}\right)} + e^{\left(j\pi\frac{V_{down}(t)}{V_\pi}\right)}\right\} \tag{1}$$

where $V_\pi$ is the MZM switching voltage, $P_{in}$ is the optical power output of LD, $f_c$ is the continuous optical wave frequency, $V_{up}(t)$ is the upper signal drive and $V_{down}(t)$ is the lower signal drive.

$X_{TX}(t)$ is given by

$$X_{TX}(t) = V_m \cos 2\pi f_m t \tag{2}$$

where $V_m$ is the RF signal amplitude, $f_m$ is the RF signal frequency, $V_{up}(t))$ and $V_{down}(t)$ of DD-MZM are given by

$$V_{up}(t) = V_m \cos(2\pi f_m t) + V_{bias} \tag{3}$$

$$V_{down}(t) = V_m \cos(2\pi f_m t + \theta) \tag{4}$$

where $\theta$ is the phase of the EPS in radians. By substituting Equations (3) and (4) into Equation (1)

$$E_{dual}(t) \approx \frac{1}{2}\sqrt{2P_{in}}e^{j2\pi f_c t}\left\{e^{(j\pi\frac{V_m}{V_\pi}\cos(2\pi f_m t)+j\pi\frac{V_{bias}}{V_\pi})} + e^{(j\pi\frac{V_m}{V_\pi}\cos(2\pi f_m t+\theta))}\right\} \tag{5}$$

is obtained. For simplification, Equation (5) can be rewritten as

$$
\begin{aligned}
E_{dual}(t) &= \frac{1}{2}\sqrt{2P_{in}}e^{j2\pi f_c t}\left\{e^{(jm\cos(2\pi f_m t)+j\pi\gamma)} + e^{jm\cos(2\pi f_m t+\theta)}\right\} \\
&= \frac{1}{2}\sqrt{2P_{in}}e^{j2\pi f_c t}\left\{e^{jm\cos(2\pi f_m t)}.e^{j\pi\gamma} + e^{jm\cos(2\pi f_m t+\theta)}\right\}
\end{aligned}
\tag{6}
$$

where $m = \pi\frac{V_m}{V_\pi}$ is the DD-MZM modulation index, and $\gamma = \frac{V_{bias}}{V_\pi}$ is the normalized bias voltage. To simplify the analysis, Equation (6) is written as

$$E_{dual}(t) = \frac{1}{2}\sqrt{2P_{in}}e^{j2\pi f_c t}\left\{A.e^{(j\pi\gamma)} + B\right\} \tag{7}$$

where

$$A = e^{jm\cos(2\pi f_m t)} \tag{8}$$

$$B = e^{jm\cos(2\pi f_m t+\theta)} \tag{9}$$

By applying Jacobi Anger expansion [31], where

$$e^{jm\cos x} = \sum_{n=-\infty}^{\infty} j^n J_n(m)e^{jnx} \tag{10}$$

and

$$e^{jm\sin x} = \sum_{n=-\infty}^{\infty} J_n(m)e^{jnx} \tag{11}$$

Equations (8) and (9) become

$$A = \sum_{n=-\infty}^{\infty} j^n.J_n(m).e^{jn(2\pi f_m t)} \tag{12}$$

$$B = \sum_{n=-\infty}^{\infty} j^n.J_n(m).e^{jn(2\pi f_m t+\theta)} \tag{13}$$

where $J_n(m)$ is the $n$th Bessel function of the first kind; therefore, Equation (6) can be stated as

$$
\begin{aligned}
E_{dual}(t) &= \frac{1}{2}\sqrt{2P_{in}}e^{j2\pi f_c t}\left\{\sum_{n=-\infty}^{\infty} j^n.J_n(m).e^{jn(2\pi f_m t)}.e^{j\pi\gamma} + \sum_{n=-\infty}^{\infty} j^n.J_n(m).e^{jn(2\pi f_m t+\theta)}\right\} \\
&= \frac{1}{2}\sqrt{2P_{in}}\left\{\sum_{n=-\infty}^{\infty} j^n.J_n(m).\left(e^{j\pi\gamma} + e^{jn\theta}\right).e^{j2\pi(f_c+nf_m)t}\right\}
\end{aligned}
\tag{14}
$$

From Equation (14), it can be seen that the DD-MZM output optical field consists of a carrier and sideband with infinite order. In accordance with the Bessel function, the proportion of carrier and sideband depends on the modulation index $m$ used. The greater $m$ is, the carrier power decreases, and more sideband orders are formed. Various forms of modulation schemes can be generated by setting parameters $\theta$ and $\gamma$.

The OSSB modulation scheme can be generated by setting the value of $\gamma = 1/2$ and $\theta = 90°$. With this arrangement, the lower sideband of order $(4n + 1)$ is suppressed, where $n = 0, 1, 2, \ldots$ [15]. The equation $E_{dual}(t)$ for OSSB modulation is given by

$$E_{dual-OSSB}(t) = \frac{1}{2}\sqrt{2P_{in}}\left\{\sum_{n=-\infty}^{\infty} j^n.J_n(m).\left(e^{j\frac{\pi}{2}} + e^{jn\frac{\pi}{2}}\right).e^{j2\pi(f_c+nf_m)t}\right\} \tag{15}$$

### 2.2. DD-MZM with Carrier Arm (CA)

The effectiveness of OSSB modulation to overcome DPF diminishes with increasing $m$; when $m$ increases, the carrier power of the DD-MZM output optical field decreases, and more sideband orders are formed. To compensate for the decreased carrier power due to increasing $m$, the DD-MZM series is added with a carrier arm (CA), as portrayed in Figure 1. The continuous optical wave signal generated by LD is shared using an optical splitter (OS); one part is used as an optical input of DD-MZM, and the other part is forwarded to CA to be recombined with DD-MZM output using an optical combiner (OC). The optical signal on DD-MZM is modulated by the RF signal with a similar configuration to DD-MZM with a detailed configuration, as shown in Figure 1.

The comparison of LD power entered into DD-MZM and the total LD power is expressed as

$$r = \frac{P_{DD-MZM}}{P_{in}} = \frac{P_{in} - P_{CA}}{P_{in}} \tag{16}$$

where $P_{in}$, $P_{DD-MZM}$, and $P_{CA}$ are the power sharing ratio, LD output power, and power input into DD-MZM and power input into CA, respectively. Thus, the DD-MZM output optical field ($E_{DD-MZM}(t)$) can be expressed as

$$E_{DD-MZM}(t) = \frac{r}{2}\sqrt{2P_{in}}\left\{\sum_{n=-\infty}^{\infty} j^n.J_n(m).\left(e^{jn\gamma} + e^{jn\theta}\right).e^{j2\pi(f_c+nf_m)t}\right\} \tag{17}$$

We assume that the optic arm length is less than 1 m. Therefore, the phase of $P_{CA}$ does not change significantly. The CA optical field ($E_{CA}(t)$) can be expressed as

$$E_{CA}(t) = (1-r)\sqrt{2P_{in}}e^{j2\pi f_c t} \tag{18}$$

The optical field output of OC ($E_{TX}(t)$) is the sum of $E_{DD-MZM}(t)$ with $E_{CA}(t)$, given by

$$\begin{aligned} E_{TX}(t) &= E_{DD-MZM}(t) + E_{CA}(t) \\ &= \sqrt{2P_{in}}\left\{(1-r)e^{j2\pi f_c t} + \frac{r}{2}\sum_{n=-\infty}^{\infty}\left(e^{jn\gamma} + e^{jn\theta}\right).j^n.J_n(m).e^{j2\pi(f_c+nf_m)t}\right\} \end{aligned} \tag{19}$$

From Equation (19), it can be seen that the optical signal spectrum of the DD-MZM output with CA comprises an optical carrier with infinite sideband order. Various forms of modulation schemes can still be generated by setting the parameter $\theta$ and $\gamma$. The magnitude of carrier power compensation of the DD-MZM output optical field that decreases because the increasing $m$ can be adjusted by regulating the value of $r$. The optical field of the DD-MZM output with CA will always have a carrier even if $J_0(m)$ is zero.

## 3. C/N Penalty of RoF Link

The DPF of the RoF link can be seen through the carrier-to-noise (C/N) penalty curve. The C/N penalty in this paper is defined by the ratio between RF power received at distance $L$ or $P_{rec}(L)$ compared to the output power at MZM modulator, and it has distance $L = 0$ or $P_{rec}(L = 0)$. The difference value between Prec ($L$) and Prec ($L = 0$) occurs because of the relative phase difference between sideband and

carrier. Therefore, the noise term is the phase noise. Moreover, the C/N penalty was also introduced by [32–34]. To calculate the C/N penalty of the RoF link, it is necessary to initially model and derive the $P_{rec}(L)$ value. The RoF link model that uses DD-MZM with CA as the E/O converter can be seen in Figure 1. The RoF link consists of DD-MZM with CA as the E/O converter, a dispersive optical fiber with $H(f)$ response and photodetector (PD) as the O/E converter.

To simplify the analysis, Equation (19) can be rewritten as

$$E_{TX}(t) = A_c e^{j2\pi f_c t} + \sum_{n=1}^{\infty} \left\{ A_{ln} e^{j2\pi(f_c - n f_m)t} + A_{un} e^{j2\pi(f_c + n f_m)t} \right\} \tag{20}$$

where

$$A_c = \sqrt{2P_{in}} \left\{ \left((1-r) + \frac{r}{2}.J_0(m).\left(e^{j\pi\gamma} + 1\right)\right) \right\},$$

$$A_{ln} = \sqrt{2P_{in}}.\frac{r}{2}.j^n.J_n(m).\left(e^{j\pi\gamma} + e^{-jn\theta}\right), \text{ and}$$

$$A_{un} = \sqrt{2P_{in}}.\frac{r}{2}.j^n.J_n(m).\left(e^{j\pi\gamma} + e^{jn\theta}\right).$$

The transfer function of the dispersive fiber link is given by [35]

$$H(f) = e^{\frac{j\pi D L \lambda_c^2 (f - f_c)^2}{c}} \tag{21}$$

where $D$ is the chromatic dispersion in ps/(nm.km). The $\lambda_c = c/f_c$ is the optical wavelength, $f$ is the frequency offset of the optical carrier, $c$ is the speed of light in a vacuum, and $L$ is the fiber length. When $E_{TX}(t)$ is transmitted through the fiber link, there is a phase difference between the optical carrier and the first-order sideband, $\phi$, given by

$$\phi = \frac{\pi D L \lambda_c^2 f_m^2}{c} \tag{22}$$

while the phase difference between the optical carrier and any random sideband is the square of the frequency range $(\pm n f_m)$ given by [36]

$$\phi_n = n^2 \phi \tag{23}$$

Therefore, the optical field arriving at the receiver $(E_{RX}(t))$ can be expressed as

$$E_{RX}(t) = A_c e^{j2\pi f_c t} + \sum_{n=1}^{\infty} \left\{ A_{ln} e^{j2\pi(f_c - n f_m)t} + A_{un} e^{j2\pi(f_c + n f_m)t} \right\}.e^{jn^2\phi} \tag{24}$$

At the receiver, $E_{RX}(t)$ is detected using a photodetector, which is a squared envelope operator, given by [2]

$$\left| E_{RX}(t) \right|^2 = E_{RX}(t).E_{RX}^*(t) \tag{25}$$

By simply taking the $f_m$ term,

$$\left| E_{RX}(t) \right|^2 = e^{j2\pi f_m t} \left\{ A_c A_{l1}^* e^{-j\phi} + A_c^* A_{u1} e^{j\phi} + A_{l1} A_{l2}^* e^{-j3\phi} + A_{u1}^* A_{u2} e^{j3\phi} + A_{l2} A_{l3}^* e^{-j5\phi} + A_{u2}^* A_{u3} e^{j5\phi} + \ldots \right\}$$
$$= e^{j2\pi f_m t} \left\{ \sum_{n=0}^{\infty} A_{ln} A_{l(n+1)}^* e^{-j(2n+1)\phi} + A_{un}^* A_{u(n+1)} e^{j(2n+1)\phi} \right\} \tag{26}$$

is obtained, where $A_{l0} = A_{u0} = A_c$.

The photodetector output ($X_{rec}(t)$) electrical current is equivalent to the real part of Equation (26) [35]; therefore,

$$X_{rec}(t) \approx \left\{ \sum_{n=0}^{\infty} A_{ln}A_{l(n+1)}^{*}e^{-j(2n+1)\phi} + A_{un}^{*}A_{u(n+1)}e^{j(2n+1)\phi} \right\} \cos 2\pi f_m t \tag{27}$$

The $X_{rec}(t)$ power is the square of the amplitude term in Equation (27) [2], so that the recovered RF signal power as the *L* function, $P_{rec}(L)$, is given by

$$P_{rec}(L) = \left\{ \sum_{n=0}^{\infty} A_{ln}A_{l(n+1)}^{*}e^{-j(2n+1)\phi} + A_{un}^{*}A_{u(n+1)}e^{j(2n+1)\phi} \right\}^{2} \tag{28}$$

where

$$\phi = \frac{\pi D L \lambda_c^2 f_m^2}{c},$$

$$A_{ln}^{*} = \sqrt{2P_{in}} \cdot \frac{r}{2} \cdot (-j^{n}) \cdot J_n(m) \cdot \left(e^{-j\pi\gamma} + e^{jn\theta}\right) \text{ and}$$

Equation (28) shows that the recovered RF signal is obtained by multiplying the *n*th order lower sideband with the (*n* + 1)th order conjugated lower sideband plus the multiplication of the *n*th order conjugated upper sideband with the (*n* + 1)th order upper sideband, where *n* is an integer. It can be seen from Equation (28) that $P_{rec}(L)$ is dependent on $\gamma$ and $\theta$. The C/N penalty is obtained by comparing $P_{rec}(L)$ with $P_{rec}(L = 0)$ and is given by [2]

$$\frac{C}{N}penalty = 10\log\left|\frac{P_{rec}(L)}{P_{rec}(L=0)}\right| \tag{29}$$

The DPF level in this paper is measured by using the DF given by [32]:

$$DF = \sqrt{\sum_{i=0}^{N} \frac{\left(\frac{C}{N} penalty_i - \overline{\frac{C}{N} penalty}\right)^2}{N}} \tag{30}$$

where $\frac{C}{N} penalty_i$ is the i-sample of the C/N penalty, $\overline{\frac{C}{N} penalty}$ is the average sample of the C/N penalty, and N is the number of samples. The lower the DF is, the smaller the effect of dispersion on $P_{rec}(L)$ or the better the system performance.

### 3.1. C/N Penalty of RoF Link with ODSB Modulation

The calculation of the C/N penalty aims to observe the DPF on the RoF link that uses DD-MZM as the E/O converter with ODSB modulation. This is done because IM on the RoF link generally produces modulated optical signals with a double sideband spectrum. Based on Equation (14), DD-MZM will generate a modulated optical signal with a double sideband spectrum only if it is set with parameters $\gamma = 1/2$ and $\theta = 180°$. The following steps are completed to calculate the C/N penalty of the RoF link with the modulated ODSB. This study focused on the effect of dispersion so that attenuation was eliminated. In addition, the attenuation at optical fiber is minimal.

(a) Calculate $P_{rec}(L)$ using Equation (28) with r = 1 (without CA), $P_{in}$ = 1 mw (0 dBm), $\gamma = 1/2$, $\theta = 180°$, *m* = 0.5, $\lambda_c$ = 1550 nm (*D* = 17 ps/(nm.km)), and $f_m$ = 60 GHz with *L* = 0, 0.1, 0.2, ... 5 km.

(b) From the obtained $P_{rec}(L)$ in (a), calculate the C/N penalty using Equation (29).

(c) Repeat steps (a) and (b) for *m* = 1, 1.5, 2, 2.5, 3, 3.5, and 4.

(d) The result of the calculation is shown in Figure 3a,b.

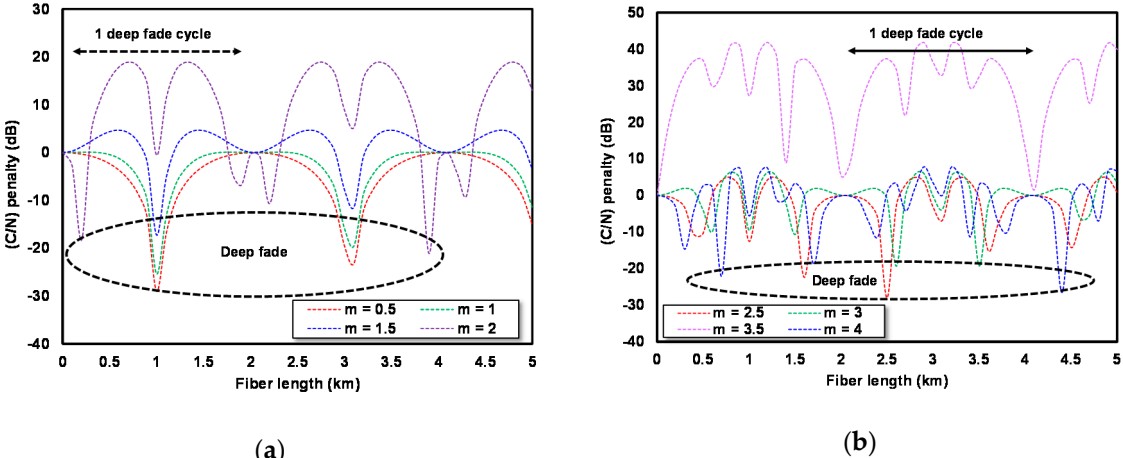

**Figure 3.** C/N penalty of the RoF link with ODSB modulation for (**a**) $m \leq 2$ and (**b**) $m > 2$.

The vertical axis in Figure 3 shows the C/N penalty in dB, whereas the horizontal axis depicts the fiber length in km. The C/N penalty of the RoF link curve with ODSB modulation at $m \leq 2$ is shown in Figure 3a, while Figure 3b shows the C/N penalty at $m > 2$. Observed from the physical meaning side, the C/N value was influenced by constructive/destructive sideband interference. Therefore, the received power followed the cosine function, and it did not strictly increase/decrease. The RoF link with ODSB modulation at $m = 0.5$, 1, and 1.5 experiences deep fade (a massive power decline) at $L = 1$ and 3.1 km; at $m = 2$, deep fade occurs at $L = 0.2$ and 3.9 km. By using Equation (30), the DF value of the RoF link is obtained with ODSB modulation at $m = 0.5$, 1, 1.5 and 2 as much as 6.5, 5.5, 4.2 and 10.5, respectively. The RoF link at $m = 2.5$, 3, 3.5 and 4 undergoes deep fade at different $L$ and has an irregular pattern. The DFs from the RoF link with ODSB modulation at $m = 2.5$, 3, 3.5 and 4 are 6.9, 5.5, 10.9, and 7.4, respectively. This shows that the level of the chromatic dispersion effect or the DPF value on the RoF link with ODSB modulation is very high.

It should be noted that several values of C/N have high positive values. This does not mean that the power of the signal was amplified. The C/N value is the ratio of the power of $P_{rec}(L)$ with $P_{rec}(L = 0)$. The C/N is positive when $P_{rec}(L)$ is higher than $P_{rec}(L = 0)$. This condition can be found when $P_{rec}(L = 0)$ has low power due to destructive interference and $P_{rec}(L)$ has high power due to constructive interference. Therefore, the C/N can be a positive or negative value.

This condition does not only apply to the irregular phase method but also to the ODSB (conventional) method. The value of $\theta$ will affect the spectrum shape of the optic signal. Moreover, the index modulation ($m$) will affect the value of $n$, and the value of $n$ will affect the number of side bands. For instance, if we apply this equation to the ODSB ($\theta = 180°$) system with parameter i.e., r = 1, $P_{in} = 1$ mw (0 dBm), $\gamma = 1/2$, $\theta = 180°$, $\lambda_c = 1550$ nm ($D = 17$ ps/(nm.km)), and $f_m = 60$ GHz, as shown in Table 1,the value of $P_{rec}(L = 0)$ is different for different values of $m$, and it was close to zero (very small). By this calculation, we use the different values of $P_{rec}(L = 0)$. Therefore, the role of $\theta$ is clear for the irregular phase as well as for ODSB.

### 3.2. C/N Penalty of RoF Link with OSSB Modulation

One of the methods used to overcome DPF is OSSB modulation. Based on Equation (14), DD-MZM can generate OSSB modulation by setting the parameters $\gamma = 1/2$ and $\theta = 90°$. The effectiveness of OSSB modulation in overcoming DPF can be seen through the C/N penalty curve of the link. The calculation of the C/N penalty of the RoF link with OSSB modulation can be completed using steps similar to ODSB modulation but using $\theta = 90°$. The results of the calculation are shown in Figure 4a,b.

**Table 1.** Calculation of $P_{rec}(L = 0)$ for ODSB modulation ($\theta = 180°$).

| Amplitude | Modulation Index ($m$) | | |
|---|---|---|---|
| | $m = 0.1$ | $m = 1$ | $m = 2$ |
| $A_{l3}$ | 0 | 0 | 0.0029 + 0.0029i |
| $A_{l2}$ | 0 | −0.0026−0.0026i | −0.0079−0.0079i |
| $A_{l1}$ | −0.0011−0.0011i | −0.0099−0.0099i | −0.013−0.013i |
| $A_C$ | 0.0224 + 0.0224i | 0.0172 + 0.0172i | 0.005 + 0.005i |
| $A_{u1}$ | −0.0011−0.0011i | −0.0099−0.0099i | −0.013−0.013i |
| $A_{u2}$ | 0 | −0.0026−0.0026i | −0.0079−0.0079i |
| $A_{u3}$ | 0 | 0 | 0.0029 + 0.009i |
| $P_{rec}(L = 0)$ dBm | $1.015 \times 10^{-8}$ | $3.41 \times 10^{-7}$ | $4.47 \times 10^{-9}$ |

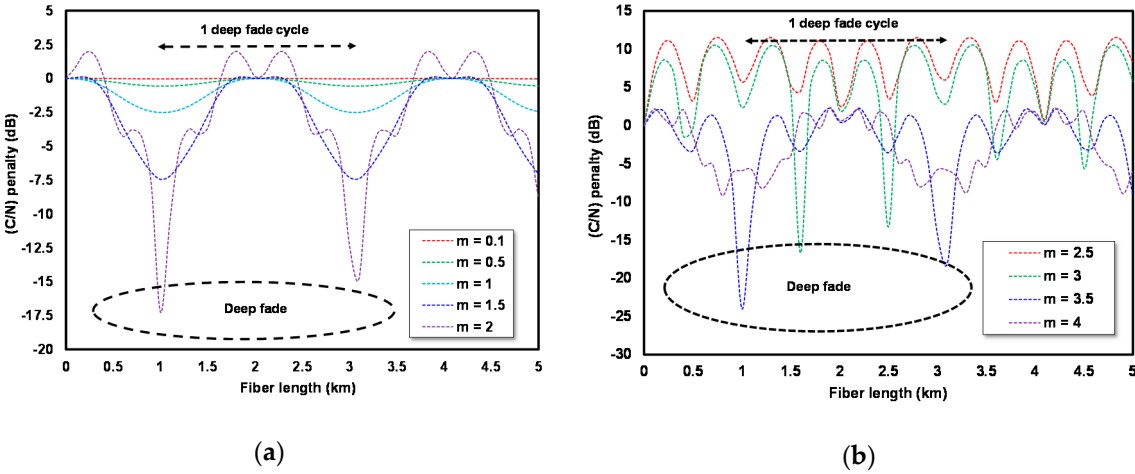

(**a**) (**b**)

**Figure 4.** C/N penalty of the RoF link with OSSB modulation for (**a**) $m \leq 2$ and (**b**) $m > 2$.

Figure 4a illustrates the curve of the C/N penalty of the RoF link with OSSB modulation at $m \leq 2$. Figure 4b shows the curve of the C/N penalty at $m > 2$. There was no deep fade on the RoF link with OSSB modulation. There was almost no difference between the C/N penalty max and C/N penalty min (Δ C/N penalty) at $m = 0.1$ (only 0.029 dB). The Δ C/N penalty at $m = 0.5$ is 0.560 dB, and it increased to 2.507 at $m = 1$. This shows that OSSB modulation could overcome DPF effectively at $m = 0.1$, and its effectiveness was reduced by increasing $m$.

A comparison of the value of the Δ C/N penalty and DF between ODSB and OSSB modulation is shown in Table 2.

**Table 2.** Comparison value of the Δ C/N Penalty and DF between the ODSB and OSSB Modulation.

| $m$ | ODSB Modulation | | OSSB Modulation | |
|---|---|---|---|---|
| | Δ C/N Penalty (dB) | DF | Δ C/N Penalty (dB) | DF |
| 0.1 | 30.111 | 6.8 | 0.022 | 0.008 |
| 0.5 | 29.046 | 6.5 | 0.560 | 0.202 |
| 1 | 25.455 | 5.5 | 2.507 | 0.930 |
| 1.5 | 21.935 | 4.2 | 7.566 | 2.764 |
| 2 | 40.013 | 10.5 | 19.264 | 4.383 |
| 2.5 | 32.922 | 6.9 | 11.517 | 3.064 |
| 3 | 25.762 | 5.6 | 27.229 | 5.629 |
| 3.5 | 41.870 | 10.8 | 26.272 | 5.207 |
| 4 | 34.368 | 7.4 | 11.525 | 3.773 |

### 3.3. C/N Penalty of RoF Link with Irregular θ

The laser diode (LD) power distribution scheme in this paper is accomplished in 4 states, i.e.,

1.  The whole optical signal power of the LD output is used as the DD-MZM input (basic configuration),
2.  75% of the optical signal power generated by LD is used as the DD-MZM input,
3.  50% of the optical signal power generated by LD is used as the DD-MZM input, and
4.  25% of the optical signal power generated by LD is used as the DD-MZM input,

While the bias point variation ($\gamma$) of DD-MZM is performed at 3 points, i.e., $\gamma = 1/4, 1/2,$ and $3/4$. Thus, all irregular $\theta$ schemes analyzed in this paper can be seen in detail in Table 3.

**Table 3.** Irregular $\theta$ scheme based on LD power distribution and variation in DD-MZM bias point.

| Bias Point | Power Ratio of $P_{DD\text{-}MZM}/P_{IN}$ | | | |
|:---:|:---:|:---:|:---:|:---:|
| | 4:4 | 3:4 | 2:4 | 1:4 |
| 1/4 | Scheme A1 | A2 | A3 | A4 |
| 1/2 | Scheme B1 | B2 | B3 | B4 |
| 3/4 | Scheme C1 | C2 | C3 | C4 |

#### 3.3.1. Irregular θ Values for Scheme A1, A2, A3 and A4

Irregular $\theta$ is the value chosen from $0°$ to $360°$ so that the RoF link produces the smallest DF. Irregular $\theta$ values are different for different schemes. Irregular $\theta$ values for scheme A1 are sought with the following stages:

(a) Calculate $P_{rec}(L)$ using Equation (28) with $r = 1$ (4:4), $\gamma = 1/4$, $m = 0.1$, $\theta = 0°$, $P_{in} = 1$ mw (0 dBm), $\lambda_c = 1550$ nm ($D = 17$ ps/(nm.km)), and $f_m = 60$ GHz at $L = 0, 0.1, 0.2, \ldots, 5$ km.
(b) Calculate the C/N penalty from the calculation results (a) using Equation (29).
(c) Calculate DF of all calculation results of (b) using Equation (30).
(d) Repeat steps (a) to (c) for $\theta = 1°, 2°, 3°, \ldots, 360°$.
(e) Find the $\theta$ value in step d), which produces the smallest DF ($DF_{min}$); this $\theta$ is called irregular $\theta$.
(f) Repeat steps (a) to (e) for $m = 0.2, 0.3, \ldots, 4$.
(g) The result of the calculation is shown in Figure 5a–d.

The examination result curve of irregular $\theta$ for scheme A1 is shown in Figure 5a. The irregular $\theta$ value on scheme A2 is searched by repeating steps (a) to (f) using $r = 3/4$, $r = 1/2$ for scheme A3 and $r = 1/4$ for scheme A4. The examination result curve of irregular $\theta$ for schemes A2, A3, and A4 can be seen in Figure 5b–d.

Figure 5 reveals that for each $m$ used, there are two values of irregular $\theta$: irregular $\theta$ I ($\theta_I$) and irregular $\theta$ II ($\theta_{II}$). The different $m$ value and different schemes produce different irregular $\theta$. The minimum DF for irregular $\theta$ I ($\theta_I$) and irregular $\theta$ II ($\theta_{II}$) is shown in Table 4.

The values of DF for each irregular $\theta$ at $0.1 \le m \le 4$ of schemes A1, A2, A3 and A4 are depicted in Figure 6. The DF values of the RoF link with an irregular $\theta$ at $m = 1$ of schemes A1, A2, A3 and A4 are 0.089, 0.119, 0.113, and 0.070, respectively, which shows that at $m = 1$, irregular $\theta$ with scheme A4 and overcome DPF most effectively on the RoF link using DD-MZM with CA, which is biased at $\gamma = 1/4$.

#### 3.3.2. Irregular θ Values for Scheme B1, B2, B3 and B4

An irregular $\theta$ value for schemes B1, B2, B3 and B4 is attained by repeating the steps in finding irregular $\theta$ values for schemes A1, A2, A3 and A4 but using $\gamma = 1/2$. The curve of irregular $\theta$ searching results for schemes B1, B2, B3 and B4, respectively, can be seen in Figure 7a–d.

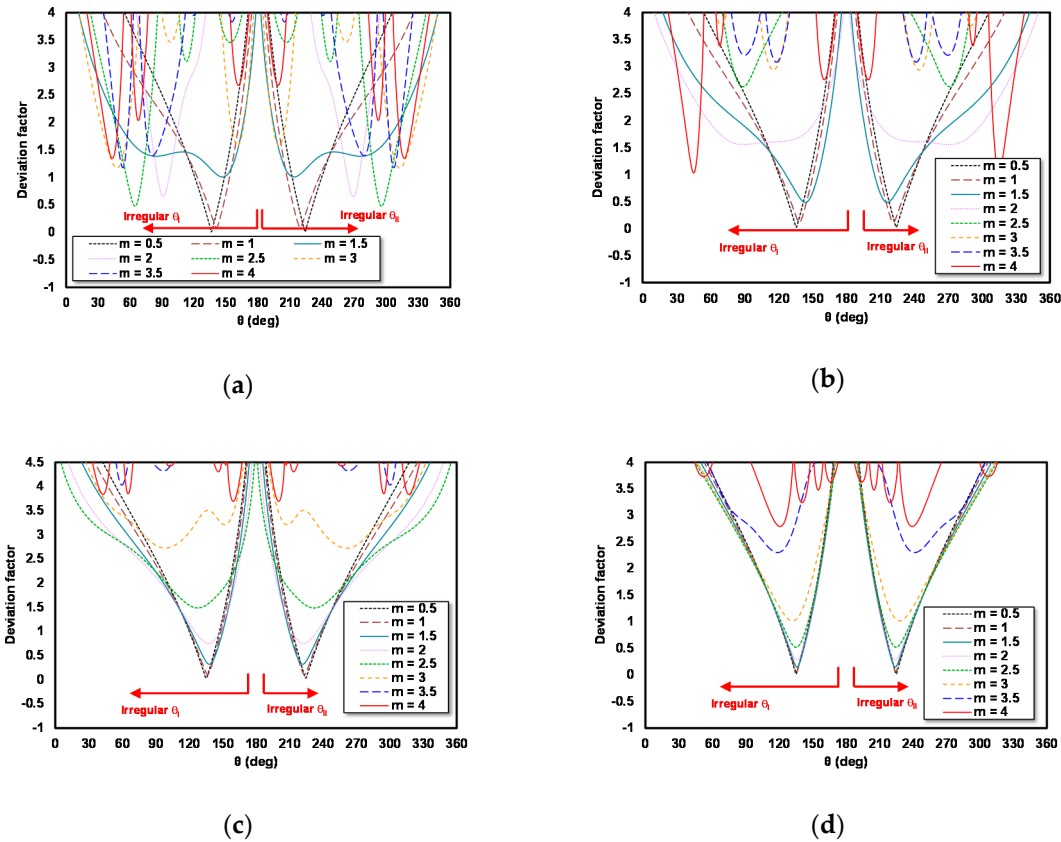

**Figure 5.** Deviation factor as the $\theta$ function for (**a**) scheme A1; (**b**) A2, (**c**) A3; and (**d**) A4.

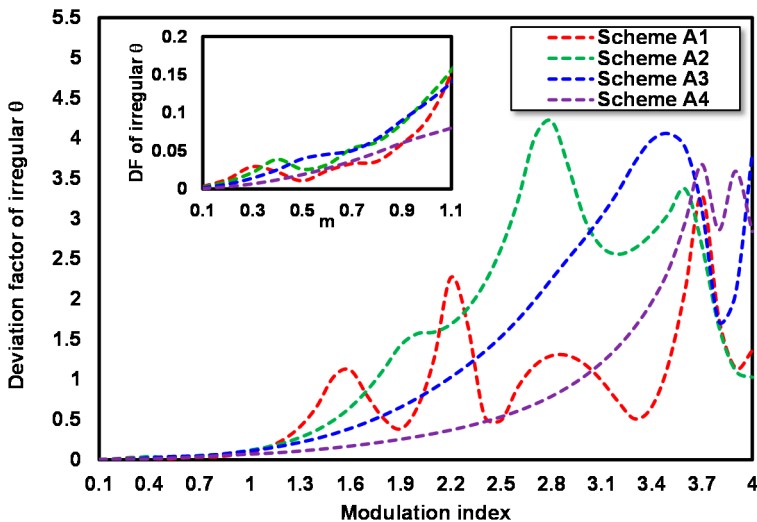

**Figure 6.** DF value for each irregular $\theta$ at $0.1 \leq m \leq 4$ of scheme A1, A2, A3 and A4.

Figure 7 reveals that for each $m$ used, there are two values of irregular $\theta$: irregular $\theta$ I ($\theta_I$) and irregular $\theta$ II ($\theta_{II}$). The different $m$ value and different schemes produce different irregular $\theta$. The minimum DF for irregular $\theta$ I ($\theta_I$) and irregular $\theta$ II ($\theta_{II}$) is shown in Table 4.

The DF values for each irregular $\theta$ at $0.1 \leq m \leq 4$ of schemes B1, B2, B3 and B4 are illustrated in Figure 8. The DF values for the RoF link with an irregular $\theta$ at $m = 1$ for schemes B1, B2, B3 and B4 are

0.093, 0.524, 0.348, and 0.131, respectively, which shows that at $m = 1$, irregular $\theta$ with scheme B1 could overcome DPF most effectively on the RoF link that uses DD-MZM with CA that is biased at $\gamma = 1/2$.

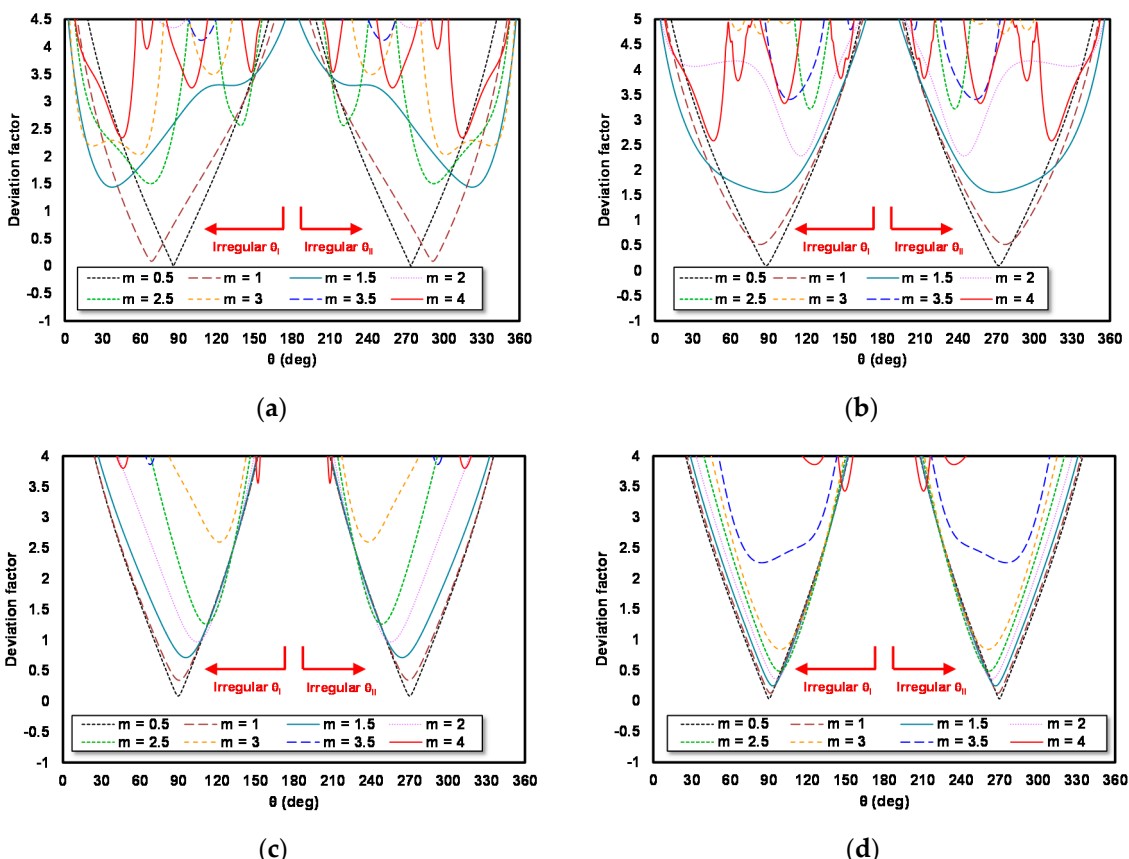

(**a**)  (**b**)

(**c**)  (**d**)

**Figure 7.** Deviation factor as the $\theta$ function for (**a**) scheme B1; (**b**) B2; (**c**) B3; and (**d**) B4.

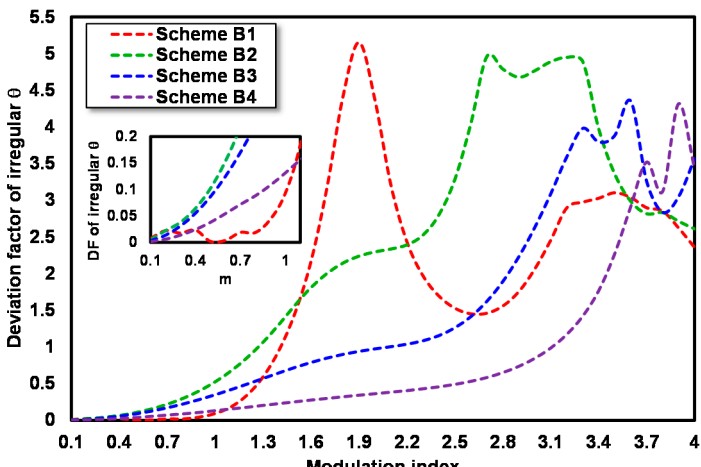

**Figure 8.** DF value for each irregular $\theta$ at $0.1 \leq m \leq 4$ of schemes B1, B2, B3 and B4.

### 3.3.3. Irregular $\theta$ Values for Scheme C1, C2, C3 and C4

An irregular $\theta$ value for schemes C1, C2, C3 and C4 is obtained by repeating the steps in finding the irregular $\theta$ value for schemes A1, A2, A3 and A4 but using $\gamma = 3/4$. The irregular $\theta$ search result curve for schemes C1, C2, C3 and C4 is shown in Figure 9a–d.

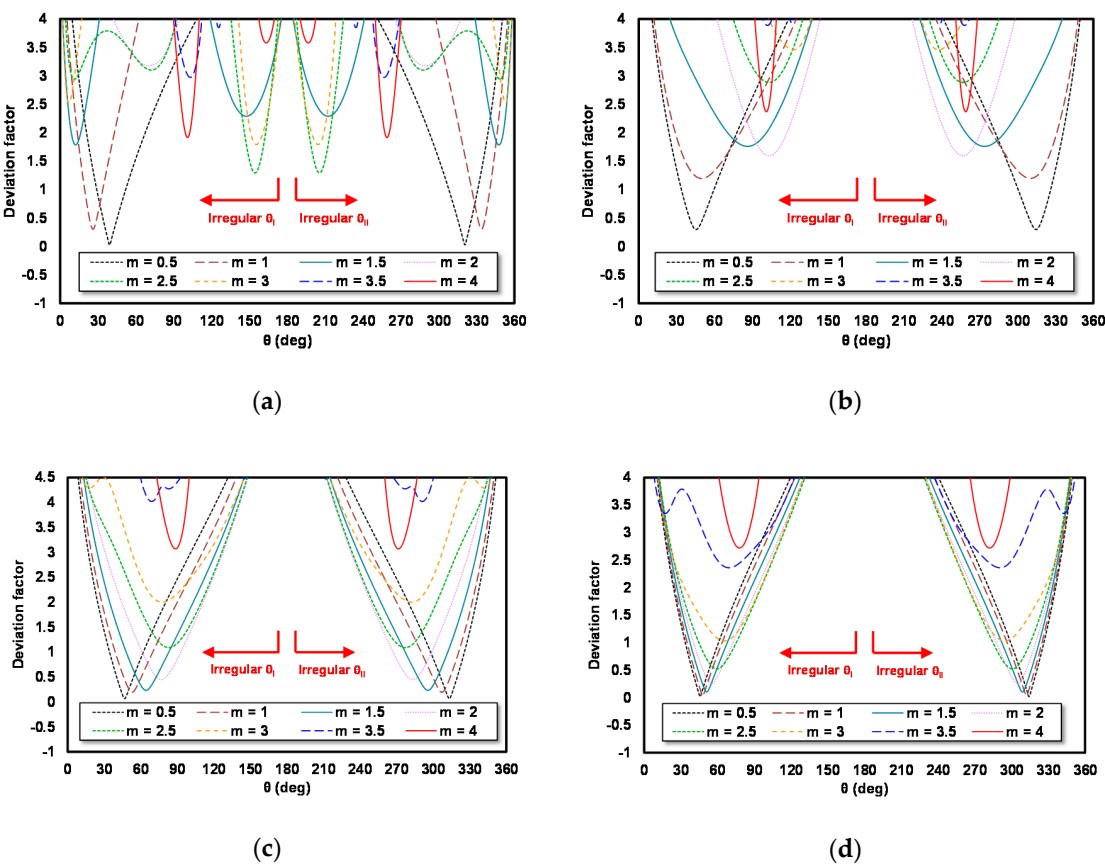

**Figure 9.** Deviation factor as the $\theta$ function for (**a**) scheme C1; (**b**) C2; (**c**) C3; and (**d**) C4.

Figure 9 shows that for each $m$ used, there are two values of irregular $\theta$: irregular $\theta$ I ($\theta_\mathrm{I}$) and irregular $\theta$ II ($\theta_\mathrm{II}$). The different $m$ value and different schemes produce different irregular $\theta$. The minimum DF for irregular $\theta$ I ($\theta_\mathrm{I}$) and irregular $\theta$ II ($\theta_\mathrm{II}$) is shown in Table 4.

The DF values for every irregular $\theta$ at $0.1 \leq m \leq 4$ from schemes C1, C2, C3 and C4 are portrayed in Figure 10. The DF values of the RoF link with an irregular $\theta$ at $m = 1$ for schemes C1, C2, C3 and C4 are 0.304, 1.196, 0.194, and 0.030, respectively. It clarifies that at $m = 1$, irregular $\theta$ with scheme C4 is able to overcome DPF most effectively at the RoF link that uses DD-MZM with CA biased at $\gamma = 3/4$.

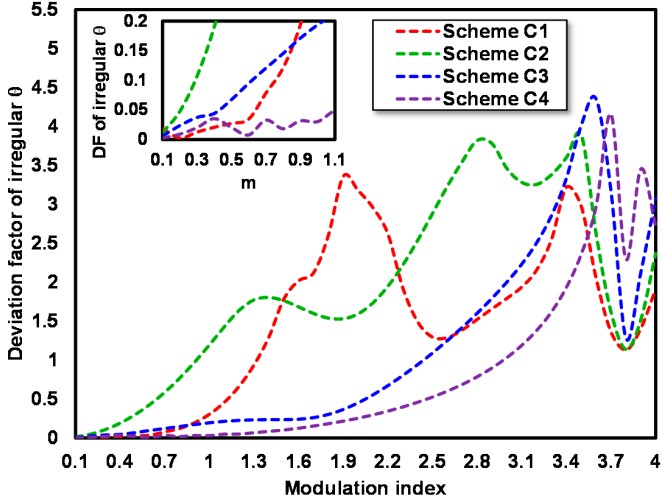

**Figure 10.** DF value for each irregular $\theta$ at $0.1 \leq m \leq 4$ of schemes C1, C2, C3 and C4.

**Table 4.** Irregular $\theta_I$ and $\theta_{II}$ for all schemes.

| $m$ | Scheme A1 | | Scheme A2 | | Scheme A3 | | Scheme A4 | |
|---|---|---|---|---|---|---|---|---|
| | $\theta_I$ (deg) | $\theta_{II}$ (deg) | $\theta_I$ (deg) | $\theta_{II}$ (deg) | $\theta_I$ (deg) | $\theta_{II}$ (deg) | $\theta_I$ (deg) | $\theta_{II}$ (deg) |
| 0.5 | 136 | 224 | 136 | 224 | 135 | 225 | 135 | 225 |
| 1 | 141 | 219 | 139 | 221 | 137 | 223 | 136 | 224 |
| 1.5 | 147 | 213 | 144 | 216 | 138 | 222 | 136 | 224 |
| 2 | 91 | 269 | 88 | 272 | 138 | 222 | 136 | 224 |
| 2.5 | 64 | 296 | 89 | 271 | 128 | 232 | 135 | 225 |
| 3 | 47 | 313 | 116 | 244 | 98 | 262 | 132 | 228 |
| 3.5 | 53 | 307 | 117 | 243 | 60 | 300 | 119 | 241 |
| 4 | 43 | 317 | 45 | 315 | 160 | 200 | 121 | 239 |

| $m$ | Scheme B1 | | Scheme B2 | | Scheme B3 | | Scheme B4 | |
|---|---|---|---|---|---|---|---|---|
| | $\theta_I$ (deg) | $\theta_{II}$ (deg) | $\theta_I$ (deg) | $\theta_{II}$ (deg) | $\theta_I$ (deg) | $\theta_{II}$ (deg) | $\theta_I$ (deg) | $\theta_{II}$ (deg) |
| 0.5 | 86 | 274 | 88 | 272 | 90 | 270 | 90 | 270 |
| 1 | 69 | 291 | 83 | 277 | 90 | 270 | 91 | 269 |
| 1.5 | 38 | 322 | 90 | 270 | 96 | 264 | 93 | 267 |
| 2 | 86 | 274 | 115 | 245 | 105 | 255 | 96 | 264 |
| 2.5 | 68 | 292 | 123 | 237 | 111 | 249 | 98 | 262 |
| 3 | 58 | 302 | 67 | 293 | 122 | 238 | 99 | 261 |
| 3.5 | 108 | 252 | 107 | 253 | 68 | 292 | 84 | 276 |
| 4 | 45 | 315 | 46 | 314 | 152 | 208 | 149 | 211 |

| $m$ | Scheme C1 | | Scheme C2 | | Scheme C3 | | Scheme C4 | |
|---|---|---|---|---|---|---|---|---|
| | $\theta_I$ (deg) | $\theta_{II}$ (deg) | $\theta_I$ (deg) | $\theta_{II}$ (deg) | $\theta_I$ (deg) | $\theta_{II}$ (deg) | $\theta_I$ (deg) | $\theta_{II}$ (deg) |
| 0.5 | 39 | 321 | 315 | 45 | 47 | 313 | 46 | 314 |
| 1 | 26 | 334 | 50 | 310 | 53 | 307 | 48 | 312 |
| 1.5 | 12 | 348 | 86 | 274 | 64 | 296 | 51 | 309 |
| 2 | 71 | 289 | 103 | 257 | 77 | 283 | 55 | 305 |
| 2.5 | 154 | 206 | 103 | 257 | 83 | 277 | 58 | 302 |
| 3 | 156 | 204 | 123 | 237 | 78 | 282 | 65 | 295 |
| 3.5 | 103 | 257 | 102 | 258 | 69 | 291 | 69 | 291 |
| 4 | 101 | 259 | 101 | 259 | 89 | 271 | 78 | 282 |

### 3.3.4. C/N Penalty of the RoF Link with an Irregular $\theta$ for All Schemes

The performance of irregular $\theta$ from all schemes in overcoming DPF can be observed through the DF value curve of irregular $\theta$ in all schemes, as depicted in Figure 11. The performance sequence of all schemes in overcoming DPF at $m = 1$ from the best to the worst is scheme C4, A4, A1, B1, A3, A2, B4, C3, C1, B3, B2 and C2. DF values of schemes C4, A4, A1, B1, A3, A2, B4, C3, C1, B3, B2 and C2 is 0.030, 0.070, 0.089, 0.093, 0.113, 0.119, 0.131, 0.197, 0.309, 0.348, 0.524, and 1.196 in order. This shows that only scheme C2 has worse performance than OSSB modulation.

To observe the effectiveness of irregular $\theta$ of all schemes to overcome DPF, the C/N penalty is calculated for all schemes at $m = 1$. The calculation is performed with $P_{in} = 1$ mw (0 dBm), $\lambda_c = 1550$ nm ($D = 17$ ps/(nm.km)), and $f_m = 60$ GHz at $L = 0, 0.1, 0.2, \ldots, 5$ km. For scheme C4, r = 1/4 and $\gamma = 3/4$ are used. $\theta_I$ for scheme C4 at $m = 1$ was 48°. For scheme A4, r = 1/4 and $\gamma = 1/4$ values are employed. $\theta_I$ for scheme A4 at $m = 1$ is 136°. r = 1 and $\gamma = 1/4$ are used for scheme A1, and $\theta_I$ for scheme A1 at $m = 1$ is 141°. Furthermore, scheme B1 uses r = 1 and $\gamma = 1/2$ values. $\theta_I$ for scheme B1 at $m = 1$ is 69°. Next, scheme A3 uses r = 1/2 and $\gamma = 1/4$ values. $\theta_I$ of scheme A3 at $m = 1$ s 137°. A2 uses r = 3/4 and $\gamma = 1/4$, and it is found that $\theta_I$ for the scheme at $m = 1$ is 139°. Scheme B4 uses r = 1/4 and $\gamma = 1/2$. $\theta_I$ for scheme B4 at $m = 1$ is 91°. For scheme C3, the values r = 1/2 and $\gamma = 3/4$ are used. $\theta_I$ for scheme C3 at $m = 1$ is 53°. Then, the values r = 1 and $\gamma = 3/4$ are used for C1. $\theta_I$ for scheme C1 at $m = 1$ is 26°. For scheme B3, the values r = 1/2 and $\gamma = 1/2$ are used. $\theta_I$ for scheme B3 at $m = 1$ was 90°. Scheme B2 uses the values r = 3/4 and $\gamma = 1/2$, and the $\theta_I$ at $m = 1$ is 83°. For scheme C2, the values r = 3/4 and $\gamma = 3/4$ are used. $\theta_I$ for scheme C2 at $m = 1$ is 50°. The calculation result of this C/N penalty is illustrated in Figure 12.

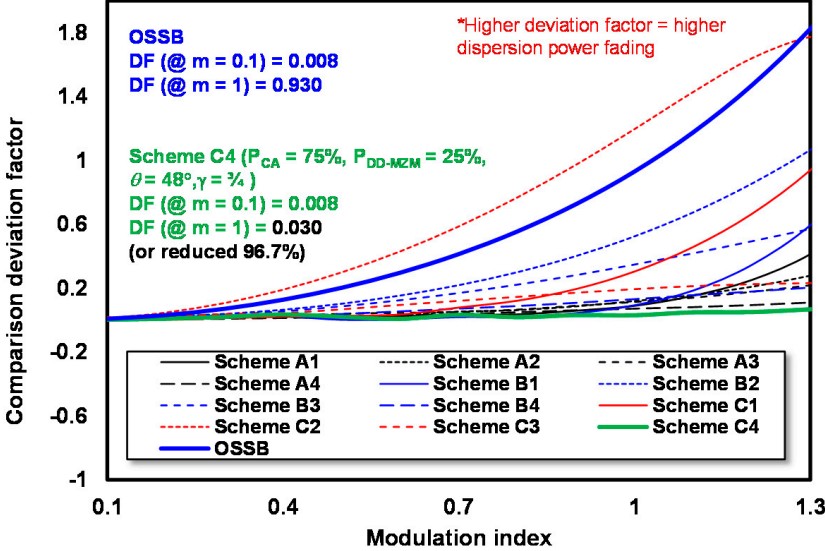

**Figure 11.** DF value of irregular $\theta$ for all schemes.

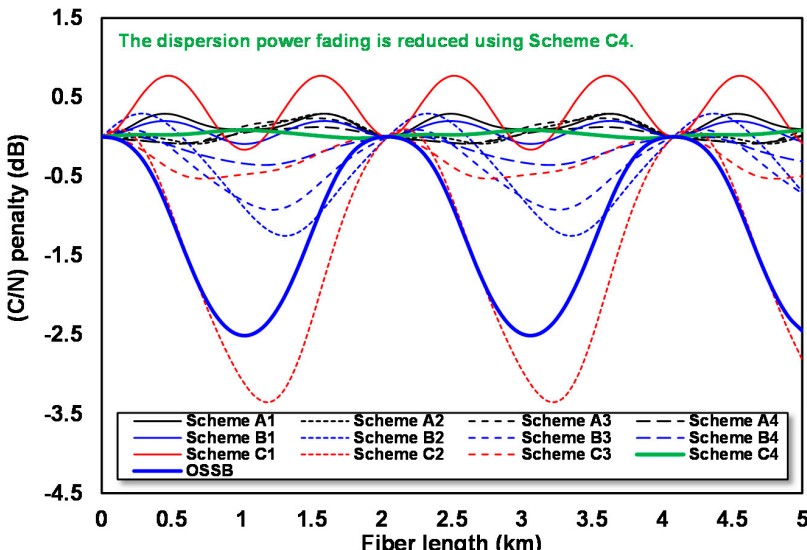

**Figure 12.** RoF links the C/N penalty with an irregular $\theta$ for all schemes at modulation index $m = 1$.

At $m = 1$, the power fluctuation that occurs on scheme C4 (a scheme with the smallest DF) is very small, where the difference between the maximum and minimum C/N penalties ($\Delta$ C/N penalty) is only 0.104 dB. The $\Delta$ C/N penalty on scheme C2 (a scheme with the largest DF) reaches 3.444 dB.

## 4. Numerical Simulation

To evaluate the performance of irregular $\theta$ in overcoming DPF, a comparison of the C/N penalty curve from the RoF link with ODSB modulation, OSSB, scheme C4 (best performing scheme) irregular $\theta$ and scheme C2 (worst performing scheme) irregular $\theta$ is carried out using OptiSystem software. An evaluation is carried out at $m = 1$ with $\lambda_c = 1550$ nm (D = 17 ps/nm.km) and $f_m = 60$ GHz. This simulation circuit is shown in Figure 13, and the parameter setting of LiNb-MZM is shown in Table 5.

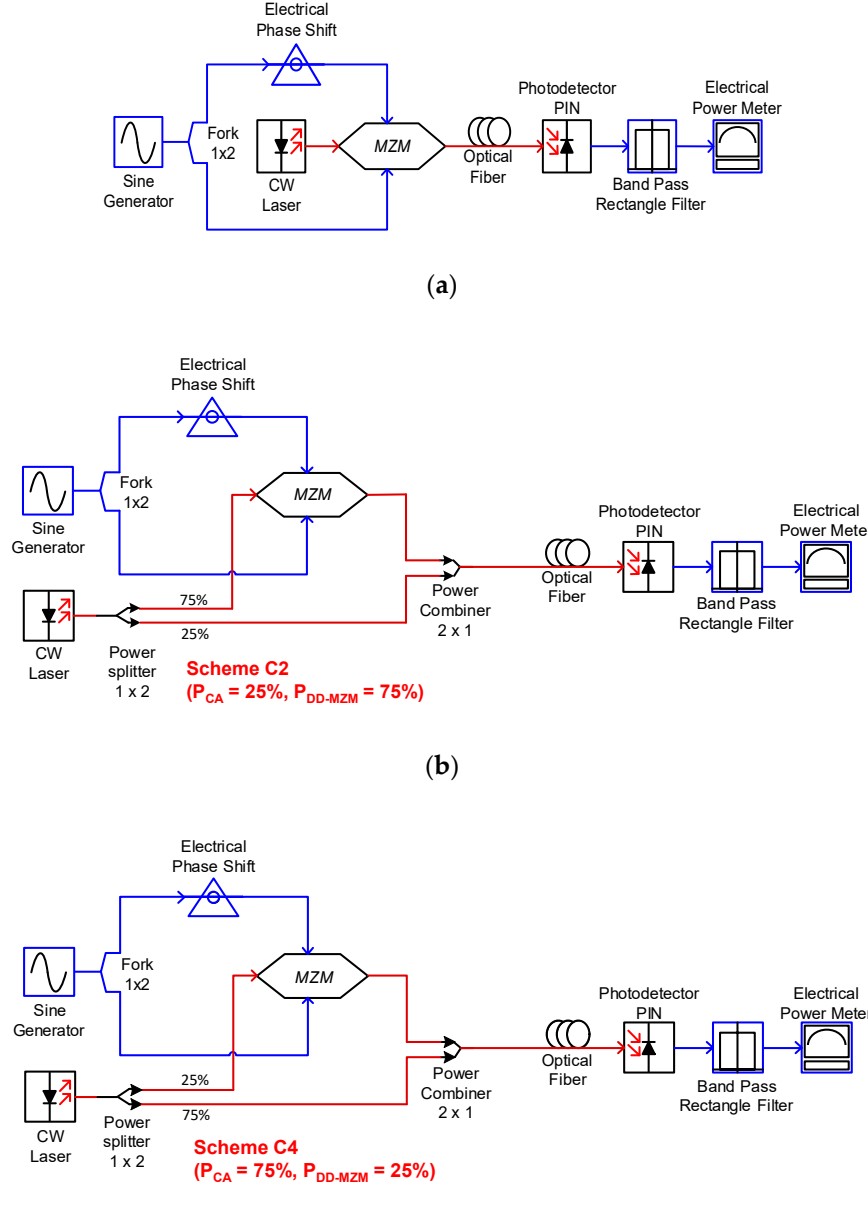

(**a**)

(**b**)

(**c**)

**Figure 13.** OptiSystem simulation system to validate irregular $\theta$ performance in overcoming DPF, (**a**) RoF link with ODSB and OSSB modulations; (**b**) RoF system with an irregular $\theta$ for scheme C2; and (**c**) scheme C4.

**Table 5.** Setting parameter of LiNb-MZM.

| Parameter | Value | Units |
|---|---|---|
| Extinction ratio | 60 | dB |
| Switching bias voltage | 4 | V |
| Switching RF voltage | 4 | V |
| Insertion loss | 0 | dB |
| Normalize electrical signal | Unchecked | - |
| Bias voltage1 | 0 | V |
| Bias voltage2 | 2 | V |

Figure 13a is a simulation system used to measure the C/N penalty of the RoF link with ODSB and OSSB modulation. The circuit consists of a sine generator, fork 1 × 2, electrical phase shift, continuous wave (CW) laser, MZM, optical fiber, photodetector PIN, bandpass rectangle filter and electrical power meter. The sine generator functions to generate the pure RF signal. The frequency of the sine generator is set to 60 GHz. Because the switching voltage $V_\pi$ used in the simulation is 4 V, the sine generator voltage $V_m$ is regulated at 1.274 V to obtain $m = 1$. The sine generator output is then duplicated using fork 1 × 2. The first fork output is forwarded to the EPS and is used as an input of the MZM upper arm. The second fork output is directly used as the input of the MZM lower arm. EPS is used to shift the RF signal phase by $\theta$. To generate ODSB modulation, the phase shift value of EPS is regulated at 180 deg and 90 deg to produce OSSB modulation. The type MZM used is LiNb-MZM. The MZM is set with parameters as depicted in Table 2. The MZM optical input is obtained from the CW laser. The frequency of the CW laser is set to 1550 nm, the power is set at 0 dBm, and the linewidth is set to 10 MHz. The output of the MZM is therefore transmitted through a single-mode optical fiber. The optical fiber is thus configured using the parameters shown in Table 6.

**Table 6.** Setting parameter of Optical Fiber.

| Parameter | Value | Units |
|---|---|---|
| User defined reference wavelength | Checked | - |
| Reference wavelength | 1550 | nm |
| Length | 0–5 | km |
| Attenuation effect | Unchecked | - |
| Group velocity dispersion | Checked | - |
| Third-order dispersion | Unchecked | - |
| Frequency domain parameter | Unchecked | - |
| Dispersion | 17 | ps/nm/km |

In this simulation, the fiber attenuation effect is ignored. At the receiver, the optical signal is detected using photodetector PIN under the parameter of responsivity = 1 A/W and dark current = 10 nA. Because the output of the photodetector consists of an electrical signal with frequencies of 0, 60, 120 GHz, etc., it is filtered by means of a bandpass rectangle filter. To obtain an RF signal at 60 GHz, the parameter filter is used with a frequency of 60 GHz, bandwidth of 10 MHz, insertion loss of 0 dB and depth of 100 dB. The power of the recovered RF signal is measured using an electrical power meter. The power measurement is performed for fiber lengths of 0 to 5 km with a step of 0.1 km. The power value in the simulation is gauged in dBm. The C/N penalty value from the simulation result is obtained by subtracting $P_{rec}(L)$ at length $L$ from $P_{rec}(L)$ at length 0.

Figure 13b is a circuit of simulations to measure the C/N penalty of the RoF link with an irregular $\theta$ on scheme C4. The components used in this circuit are approximately similar to the simulation circuit used to measure the C/N penalty of the RoF link with ODSB and OSSB modulations. Given that the DD-MZM optical input on scheme C4 is only 25% of the LD total power, the LD power in this circuit is first divided into 4 parts using a 1 × 4 optical splitter (OS). Therefore, 1 OS output is used as the MZM optical input, and 3 OS outputs are combined using the 3 × 1 optical combiner (OC). The output of 3 × 1 OC is then recombined with the output of MZM by means of 2 × 1 OC. For scheme C4, the bias voltage 2 of DD-MZM is regulated at 3 volts, and the value of the phase shift of EPS is set at 48°. The power measurement in this simulation is identical to the measurement on links with ODSB and OSSB modulations.

Figure 13c illustrates a simulation circuit to measure the C/N penalty of the RoF link with an irregular $\theta$ for scheme C4. The components used in this circuit are no different from the simulation circuit employed for measuring the C/N penalty of the RoF link with an irregular $\theta$ for scheme C2. However, this simulation uses MZM optical input by 25% from the LD total power; hence, MZM is given an input of 3 × 1 OC output. An OS output is then recombined with the output of MZM using 2 × 1 OC. For scheme C2, bias voltage 2 of MZM is also set at 3 V. The comparison result between

the mathematical model and simulation model has very good agreement, which shows the validity of the proposed method, as shown in Figure 14.

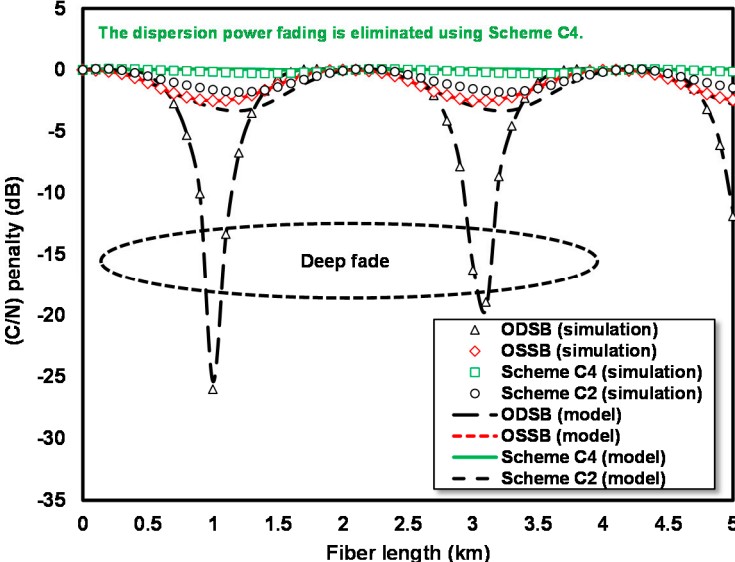

**Figure 14.** C/N penalty for the RoF link with ODSB and OSSB modulations, scheme C4 irregular $\theta$ and scheme C2 irregular $\theta$ at $m = 1$.

The C/N penalty of the RoF is linked with ODSB modulation, and both the calculation result and simulation show a deep fade at $L = 1$ and 3.1 km. However, deep fade does not occur in the C/N penalty of the RoF link with OSSB modulation, although there is still a $\Delta$ C/N penalty of 2.5 dB. The $\Delta$ C/N penalty in the C/N penalty of the RoF link curve with an irregular $\theta$ for scheme C4 was very small.

## 5. Conclusions

The dispersion power fading in RoF communication is successfully reduced using an asymmetric carrier divider with an irregular RF phase on the DD-MZ modulator. The minimum DF is obtained when the $P_{IN}$ is separated as 75% for $P_{CA}$ and 25% for $P_{DD-MZM}$ with an irregular RF signal of $\theta = 48°$ and bias point value of $\gamma = 3/4$. As the result, with the same power as OSSB, this proposed structure produces DF at $m = 0.1$ and $m = 1$ value are 0.008 and 0.03, or it can decrease the DF of 96.7% at $m = 1$. The mathematical model and simulation model have very good agreement, which validates the proposed method. The proposed scheme is more suitable for external modulation. It can be applied for applications such as supporting 5G communications or supporting IEEE 802.11 wireless network. It should be noted that the limitation at the implementation, such as the use of single-mode fiber optics, power splitter, power combiner, and polarization controller to make the polarization is constant. There are several suggestions for future research. First, the validation can be improved in future studies through measurements. Moreover, information signal analysis and bit error rate (BER) calculations can be potentially included. Lastly, iterations on power divider can be made more detailed and general.

**Author Contributions:** Conceptualization, G.W. and F.U.; methodology, G.W.; software, F.U.; validation, G.W., P.S.P. and T.F.; formal analysis, F.U.; investigation, T.F.; resources, F.U.; data curation, T.F.; writing—original draft preparation, F.U.; writing—review and editing, G.W.; visualization, T.F.; supervision, P.S.P.; project administration, G.W.; funding acquisition, G.W. All authors have read and agreed to the published version of the manuscript.

**Funding:** This research was funded by QQ Project Grant (Universitas Indonesia), grant number NKB-0299/UN2.R3.1/HKP.05.00/2019.

**Conflicts of Interest:** The authors declare no conflict of interest.

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
