# Peer review of "Asymmetric Carrier Divider with an Irregular RF Phase on DD-MZ Modulator for Eliminating Dispersion Power Fading in RoF Communication"

_photonics, doi:10.3390/photonics7040106_

Round 1

Reviewer 1 Report

In this paper, the Authors propose a method for mitigating impairments due to chromatic dispersion (CD) during signal propagation in optical fibers. To this regard, the beneficial impact of an unbalanced split of the optical signal in a dual-drive Mach-Zehnder modulator (DD-MZM) is demonstrated. Additionally, the Authors show, through numerical simulations, how a technique already employed in wireless communications against fading, like irregular RF phase, can be successfully employed in optical communications. I agree with publishing this paper, although after minor revisions.
A) The main limitation of this work is that analysis has been performed on optical signals modulated by RF sine waves. What if signals with a band are considered? In particular, which is the impact of the irregular phase technique, if phase-modulated signals are employed? Please comment on this, with proper references;
B) In Eq. 20, where the fiber response is reported, it would be beneficial for the reader to define the relation on fc and the central wavelength;
C) It is not clear the definition of the carrier-to-noise ratio (C/N), since it looks like the ratio between the optical power at the input of the fiber and the power after propagation through it, without utterly involving any noise term;
D) In Sect. 3.1, line 245, it is reported that "destructive interference is possible even at zero distance". I think this sentence should be better motivated, not simply referring to Eq. (27) since, from that, it looks like interference is possible at zero distance (i.e. zero CD) only if irregular phase theta comes into play. Please explain better the role of theta in this.
E) Clearly, theta must depend on the fiber span length; what if the path the signal takes changes, e.g., due to networking routing rules?
F) Finally, I find the exposition somehow redundant (the same information repeated several times in the same way, the same acronyms defined twice...), although it is very clear. However, I suggest more attention in equations formatting (e.g., use Microsoft Equation tool, if you are using Office). The Authors should check more carefully the text for typos, like in line 179 "carrier-to-nose (C/N)", line 226 "dan" for "and".

Reviewer 2 Report

The authors describe a system for high-speed radio-over-fiber (RoF) communication and claim to use a new method to combat the chromatic dispersion of optical fibers through an innovative asymmetric carrier splitter with an irregular high-frequency phase on the Mach-Zehnder modulator with dual drive. I see the application of such a system mainly in wireless networks (i.e. IEEE 802.11 or 5G). Considering the fact that radio- over-fiber (RoF) communication is of great interest in wireless communication and that the asymmetric carrier splitter may be important to avoid of fiber dispersion power penalty, I recommend the publication of this paper after several classifications and improvements:

1) It is mentioned in the introduction that there are several methods to overcome the power penalty induced by the dispersion of optical fiber, but only one method for carrier shift and one method for optical carrier suppression are mentioned. It will be useful for the reader to list some other solutions:

(e.g. M. A. Ilgaz, et al., “A flexible approach to combating chromatic dispersion in a centralized 5G network”, Opto-Electronics Review, 2020, vol. 28, No. 1, pp. 35-42)

optical dispersion compensation modules as fiber Bragg grating and dispersion compensation fibers (e.g. Ranzini, S.M.; Da Ros, F.; Bülow, H.; Zibar, D. Tunable Optoelectronic Chromatic Dispersion Compensation Based on Machine Learning for Short-Reach Transmission. Appl. Sci. 2019, 9, 4332.)

- single passband microwave photonic filter (e.g. Li, L.; Yi, X.; Song, S.; Chew, S.X.; Minasian, R.; Nguyen, L. Microwave Photonic Signal Processing and Sensing Based on Optical Filtering. Appl. Sci. 2019, 9, 163.)

2) In Figure 1 I propose to mark the ordinate of the subimages 1 to 6 with P_optical and subimage 7 with P_RF. Please additionally correct subimages 4, 5 and 6, where "f_c+f_m" covers the presented frequency component. I also suggest to add another subimage to represent the RF signal of X_TX.

3) The polarization of the laser diode is not taken into account, but is always a very important factor in practical experiments, since the Mach-Zehnter modulator and the additional arm are polarization dependent. A comment on polarization will be more than welcome in the paper.

4) The length of the additional optical arm is not taken into account. It may change the phase of P_CA and there may be constructive or destructive interference at the output of the optical combiner. In this paper only the case of constructive interference is considered. A comment on this will be more than welcome in the paper.

5) This is a very relevant area of research, but it would be more useful for the community if the work were extended by an experimental implementation of the asymmetric carrier splitter with an irregular high-frequency phase on the Mach-Zehnder dual drive modulator. Some experimental demonstrations should be provided to support the proof of concept, at least for the preliminary stage. The lack of experimental validation is a major weakness.

Reviewer 3 Report

    The authors proposed a DPF-compensating scheme in RoF by adjusting the power ratio of the modulating signal over the carrier signal, and the phase of the RF message. This paper is generally well presented, but the simulation results are not solid enough for fully describing the system performance. My comments are listed as follows:

General Remark

  • Does the proposed scheme belong to either ODSB or OSSB? Both of them have a specific RF phase for the low-arm signal entering the MZM, but the RF phases of the proposed scheme are not fixed.
  • Which one is the proposed scheme more suitable for, the direct or external modulation? Please verify their applications and limitations.
  • Please give the equation of Edual(t) for the OSSB modulation given the conditions in lines 155-156.
  • Please explain why C/N penalty varies with the fiber length in Figures 3 and 4. Furthermore, most paragraphs describing Figures from 6 to 12 simply mention the parameter values plotted in these graphs. I would expect more interpretations of the physical meaning of the variations of these parameter values. Especially, most curves are oscillating and not strictly increasing/decreasing.
  • Up to 16 cases are analyzed in this paper, but the derivation of the optimal values of q and power ratio should be further investigated. In other words, the minimum DF might not be reached when the power distribution of Pca and PMZM is 3:1, as only four cases of different ratios are tested in this paper. An analysis of C/N penalty in the y-axis and power ratio in the x-axis would help. The contribution of this paper is not adequately shown as the analysis is simply limited to some user-defined cases.
  • The transparency of the proposed scheme seems to be limited as the optimal q and power ratio should be searched again whenever the system setup is changed. The wrong assignments of the values of these two parameters may result in worse performances than the ones of the pure OSSB. Please comment that and discuss how to find the optimal setup efficiently.

Minor Remark

  • Some keywords are not defined, and generally, the terms in Keywords should  be also shown in Abstract.
  • In lines 71, there are too many references without review and classification.
  • The symbols and parameters in equations should be italic.
  • In (6) and (25), there are two equations for a single number. Please consider to combine them or mark them as distinct numbers.
  • In line 212, “gamma” and “theta” should be denoted as Greek symbols.

Round 2

Reviewer 2 Report

My main concerns about the manuscript are answered and corrected. The paper can accept as it stands.

Reviewer 3 Report

The responses to common 1 and common 3 are the same. Please revise them. Other responses are acceptable.
